# Measurement-Based Care in Youth: An Opportunity for Better Clinical Outcomes?

**DOI:** 10.3390/healthcare12090910

**Published:** 2024-04-27

**Authors:** Roberta Frontini, Catarina Costa, Sílvia Baptista, Constança do Carmo Garcia, António Vian-Lains

**Affiliations:** 1Clínica de Neurociências e Saúde Mental, Hospital Cruz Vermelha, 1500-048 Lisboa, Portugal; catarina.costa@hcvp.com.pt (C.C.); silvia.baptista@hcvp.com.pt (S.B.); antonio.lains@hcvp.com.pt (A.V.-L.); 2Faculdade de Ciências Humanas, Universidade Católica Portuguesa, 1649-023 Lisboa, Portugal; s-cchgarcia@ucp.pt

**Keywords:** assessment, measurement-based care, youth

## Abstract

Measurement-based care (MBC) is a procedure in which systematic and routine assessments are performed. Through this practice, clinicians can verify the progress of the symptomatology of the patient and adapt the appointments and the intervention to the current symptoms. Studies have reflected on the importance and the benefits of this type of procedure in the adult population, and have shown positive results. However, there is a lack of evidence concerning the remaining populations. Regarding youth, for instance, few articles have evaluated the benefits of using this procedure in clinical practice. However, research focused on this topic has revealed positive results, especially when clinicians were loyal to the MBC procedures. Still, further research is needed. This letter aims to share the methodology used by our multidisciplinary team, composed of psychologists and psychiatrists, in a clinical context at the *Hospital Cruz Vermelha*, Lisboa, applied to the adult population; the objective is to share and discuss some alterations that could be made to our evaluation protocol to enable the same to be used with the youth population. We believe that implementing MBC for youth is crucial for several reasons, including enhanced treatment efficacy, more personalized treatment, a reduced reliance on subjectivity, and empowerment not only of patients but also families.

## 1. Measurement-Based Care (MBC)

According to the American Psychological Association [1], Measurement-Based Care (MBC) is a systematic assessment performed during the therapeutic process. It provides a better understanding of mental disorders, collecting data about the symptoms at the time of assessment; namely, the severity, frequency, and impact on the functioning of the individual [2]. Additionally, MBC supports the procedure of engaging the patient in the decision-making process and in discussions about the overall treatment [1,3]. In general, this assessment is based on the patient’s self-report, considering their perspective on symptomatology and functioning [1,3]. This evidence-based practice has some key components, such as collecting data routinely, sharing the results with the patient, and acting in accordance with the data [1]. The benefits of using this clinical process have been well described in the literature; namely, improved response rates [4], decreased response time [5], decreased costs associated with the therapeutic process [6], higher levels of engagement [7], enhanced overall functionality [8,9], and increased quality of therapeutic alliance [10]. 

Moreover, the MBC has been shown to have transtheoretical and transdiagnostic relevance across clinical settings [2]. A recent theoretical review of MBC in psychiatry identified additional advantages of MBC when compared to the usual care, such as identification of the patients who are improving or deteriorating; improvement of role functioning, satisfaction with care, quality of care, and quality of life; communication between providers and patients; improvement of collaborative care efforts among providers; improvement of the accuracy of clinical judgment; enhanced individualized treatment; and the viability of implementing MBC on a large scale [11]. Besides the benefits, Aboraya and colleagues also listed some potential barriers, particularly related to the fact that the measures are time-consuming (the most commonly cited drawback by psychiatrists), ratings produced by the measures might not always be clinically relevant, and administering rating scales might interfere with establishing rapport with patients; additionally, there is the perception that the measures are not more useful than clinical assessment; the perception that MBC is over-systematizing and depersonalizing; the limited formal training (included in the top two barriers for residents and faculty); the lack of protocols and training manuals; the lack of consensus as to which instrument to use for a particular disorder; the absence of a requirement to use MBC—particularly since few work settings require MBC; the lack of incentives to use MBC; and the complexity of patients with multiple overlapping comorbidities.

## 2. The MBC Protocol Applied at the *Hospital Cruz Vermelha*

In the Clínica de Neurociências e Saúde Mental of *Hospital Cruz Vermelha* (Lisboa, Portugal) this practice is applied to adults (patients 18 years or older) who attend psychology or psychiatry appointments or who are forwarded from other hospital services. 

Initially, a more extensive version of the protocol is applied (Time 0). The average application time has been one hour. This protocol includes self-report questionnaires, such as the Patient Health Questionnaire-9 (PHQ-9) [12,13] to assess the severity of depressive symptoms, the General Anxiety Disorder questionnaire (GAD-7) [14,15] to assess anxiety, the Center for Epidemiologic Studies Anxiety Scale (CESA) [16] to assess anxiety, the WHO Disability Assessment Schedule (WHODAS 2.0) [17,18] to assess patient functionality, the Sick-Control-One stone-Fat-Food (SCOFF) [19] to assess eating behavior, the DSM-5-TR Self-Rated Level 1 Cross-Cutting Symptom Measure—Adult [20] to assess somatic symptoms, psychotic symptoms, personality disorders, bipolar disorders, and obsessive-compulsive disorder, the Adult Self Report Scale (ASRS–V1.1) [21,22] to assess hyperactivity disorder and attention deficit, the Pittsburg Sleep Quality Index (PSQI-PT) [23,24] to assess the quality of sleep, and the Alcohol, Smoking and Substance Involvement Screening Test (ASSIST) [25,26] to assess the involvement of substance use. This protocol also includes questions that aim to evaluate symptomatology related to panic disorder, obsessive-compulsive disorder, post-traumatic stress disorder, mania, and personality traits. Other relevant information is collected during the clinical process, such as adherence to psychiatric pharmacological treatment (if applied) (questionnaire adapted [27]), presence of comorbidities of physical illnesses (questionnaire adapted [28]), family history of mental illness, and previous use of specialty psychology and/or psychiatry services. To complete the protocol, a screening of cognitive dysfunction is performed through a tablet utilizing the program THINC-IT [29,30]. In the case of severe difficulties, or if the patient is 65 years or older, this test is replaced by the Montreal Cognitive Assessment (MoCA) [31,32]. 

In order to automate the scoring and interpretation of the results, our team has developed a software tool, Psymap 1.1, which, based on the individual scoring results, automatically generates a clinical note for the patient’s electronic health records (EHRs) and an individualized visual report for patient education. This greatly contributes to engaging patients in discussions about treatment strategies and goals. The software tool also helps attenuate the burden of completing EHRs, thus optimizing appointment time, which leads to an increase in the response capacity of the mental health service.

The protocol is applied at the time of the first appointment. The Psymap software automatically scores the scales and presents the results of the clinician. This allows for including a broad collection of symptoms covering different areas of functioning, thus facilitating a differential diagnosis and the identification of possible comorbidities during the patient’s first visit without overburdening the mental health professional. At the time of each follow-up visit (Time 1 and subsequent), a shorter version of the protocol is applied, comprising the assessments that are more sensitive to change. This smaller questionnaire has an average application time of between 10 and 15 minutes and is usually completed by the patient in the waiting room. It is comprised of the Patient Health Questionnaire-9 (PHQ-9) [12,13], the General Anxiety Disorder questionnaire (GAD-7) [14,15], the Center for Epidemiologic Studies Anxiety Scale (CESA) [16], and the WHO Disability Assessment Schedule (WHODAS 2.0) [17,18]. Every six months, the full protocol is re-applied.

## 3. MBC in Youth

According to the National Survey of Children’s Health, 16.5% of youth suffer from at least one mental disorder [33]. The literature suggests that 11.2% of the youth population meets the criteria for mood disorders, 8.3% for anxiety disorders and 9.6% for behavioral disorders [34]. However, mental illness in youth largely remains untreated or unrecognized, leading to an increase in social exclusion, discrimination, and educational difficulties [35,36,37,38,39]. Therefore, it is extremely important to enhance the psychological assessments applied to this population. Thus, it could be important to use the same method applied to the adult population when applying an MBC protocol to youth; this could be done in different contexts such as schools, minimizing the potential impact on public health [40]. 

As in the other areas, the benefits of MBC are less studied in youth when compared to the adult population. However, there is a growing body of literature supporting its implementation in the youth population as well. According to Jensen-Doss and colleagues [41], the MCB assessment is well suited for working with youth. Both the Transdiagnostic Treatment of Emotional Disorders in Adolescents [42] and Parent–Child Interaction Therapy (PCIT) [43] resort to MBC to identify treatment goals and to review the progress of the patients. The randomized controlled trial performed by Bickman and colleagues [44], found benefits of MBC in a sample composed of youth participants, such as faster improvements, enhanced monitoring of the therapeutic relationship, and adjustment of the clinician’s approach according to the patient’s needs. Moreover, in a second randomized controlled trial, the results were only positive, when the clinicians were loyal to the MBC procedures [45]. 

The literature suggests that MBC in youth, when shared with stakeholders (such as developers, researchers, and consumers), optimizes the clinical outcomes through supported clinical decision-making [46]. It can also be applied to young patients as well as caregivers, supervisors, and administrators [47], and may be applied in different contexts, such as schools.

Thus, the evidence supports that MBC is useful in improving mental health outcomes in the youth population [40]. However, despite all the evidence in its favor, clinicians rarely engage in this practice [40] with an estimated adherence by mental health professionals of only 20% [6]. Table 1 summarizes some studies that support the use of MBC in clinical practices with the youth population.

## 4. MBC Protocol When Applied to a Youth Population 

When developing a protocol for psychological assessment to be applied to a youth population, it is important to consider a variety of methods including interviews, psychometric tests, and observation; all these methods must be applied to the patient and, if justified, to significant figures such as caregivers or teachers [48]. It is also important to select short-form questionnaires, reducing fatigue levels and increasing the likelihood of adherence, whether by the person or by the informants [11].

All the questionnaires must be carefully selected, validated for the intended population (considering aspects like age and nationality), present good psychometric qualities, and be sensitive to change, enabling the systematic assessment [11].

Before starting the assessment, it is important to ensure that informed consent is obtained [49]; this must be provided by the patient or the legal guardians, depending on the age and the local laws. Moreover, and even when the informed consent form is signed by their legal guardians, it is of utmost importance to ask the child or adolescent if they want to participate. It is also important to clarify the limits of confidentiality [50]. 

During the psychological assessment, it is important to guarantee a sensitive and empathetic approach and a comfortable environment, with the use of appropriate language and avoiding technical nomenclature, granting the understanding and comprehension of the instructions and questions [51]. The active participation of the young patients must be encouraged by providing a safe place where they feel heard. To guarantee the security of the patients, it is extremely important to assess the safety risks (suicide risk, self-harm, abuse, or violence) [52]. 

When contemplating the results obtained, it is crucial to proceed with contextualization, considering the developmental norms for different age groups [53], and the cultural and ethnic differences that might influence the answers to the assessment [54]. It is also important to give feedback to the patient or other relevant figures who might be concerned, discuss the results, and provide post-assessment care, if necessary [1]. 

Along with these important considerations, there is a rising body of evidence supporting the implementation of MBC in youth mental health care [55]. Several articles have tried to discuss the possible components of the MBC protocol in youth. A systematic review and meta-analysis reviewed 12 studies and concluded that there were promising early findings to support MBC in the youth population [56]. In fact, the National Institute of Mental Health, in consultation with the Welcome Trust [57] and other funders of mental health research, acknowledged a minimum list of ideal instruments to use in mental health settings. For research participants younger than 18, they suggest assessing the age; the sex at birth; the mental health domains, using the DSM-5 Self-Rated Level 1 Cross-Cutting Symptom Measure Parent/Guardian Report and the Youth Self Report [20]; and the levels of anxiety and low mood, using the Revised Children’s Anxiety and Depression Scale (RCADS-25) (both the Parent/Caregiver Report and the Youth Self Report) [58,59].

## 5. Conclusions

It is our understanding that implementing MBC in youth is vital for several reasons: it allows the enhancement of treatment efficacy, provides a more personalized and individualized treatment, reduces the reliance on subjectivity, and empowers not only patients but also their families. 

As in other areas, the benefits of MBC are less studied in youth compared to the adult population. However, there is a growing body of literature supporting its implementation in the youth population as well. To sum up, we believe the alterations necessary to make it suitable for young individuals should consider:Tailored Assessments for Youth:
oPsychological assessments for youth should be tailored to their developmental stage and needs.oConsideration should be given to age-appropriate language and comprehension levels.oUse of short-form questionnaires to minimize fatigue and increase adherence.oTake into consideration the informed consent process and ensure the child or adolescent’s willingness to participate even if legal guardians provide consent.oEncourage active participation from young patients to ensure they feel heard and understood.
Consider Safety Assessment:
oAssess safety risks including suicide risk, self-harm, abuse, or violence to ensure the well-being of the patient.
Contextualization of Results:
oContextualize assessment results considering developmental norms, cultural, and ethnic differences.oProvide feedback to the patient and discuss assessment results comprehensively.oProvide and offer post-assessment care if necessary.

Thus, and considering the aforementioned information, with this letter, we would like to increase the debate around the use of MBC and increase the discussion regarding the best questionnaires/assessment tools to be used in the youth population.

## Figures and Tables

**Table 1 healthcare-12-00910-t001:** Summary of studies that support the use of MBC in clinical practice with youth.

Year	Complete Reference	Aims	Main Findings
2022	McLeod et al. [47]	To explore the utility of the Measurement-Based Care (MBC) approach for youth. To specify ways to enhance the utility of MBC, considering implementation in real-world settings and addressing associated practical and ethical challenges.	Review of existing evidence on the effectiveness and utility of MBC for youth. Identification of gaps in knowledge or areas where the implementation of MBC can be improved. Suggestions for improving clinical practice related to MBC for youth, including recommendations for future research and guideline development.
2020	Jensen-Doss et al. [41]	To explore the use of Measurement-Based Care (MBC) as a tool for improving clinical practice in youth mental health settings. To investigate how MBC can be applied at both the clinical and organizational levels to enhance care delivery and resource allocation. To provide insights into the potential benefits and challenges of implementing MBC in youth mental health settings.	MBC can effectively inform treatment decisions and monitor progress in youth mental health care. Implementing MBC at the organizational level can lead to improvements in the quality-of-care delivery and resource allocation.
2019	Lyon et al. [40]	To investigate the implementation of a digital feedback system designed to support Measurement-Based Care (MBC) by school-based mental health clinicians. To assess the effectiveness of the digital feedback system in facilitating the use of MBC tools and improving clinical decision-making in school-based mental health settings. To explore the feasibility and acceptability of integrating digital technology into routine clinical practice for youth mental health.	The digital feedback system effectively supported the implementation of MBC by school-based mental health clinicians. Clinicians reported increased use of MBC tools and greater confidence in their clinical decision-making processes. The digital feedback system facilitated communication between clinicians and enhanced collaboration within multidisciplinary teams. Improved outcomes for students receiving mental health services in school-based settings may be observed as a result of more systematic and data-informed care. Recommendations for the successful integration and sustained use of digital feedback systems in school-based mental health practice are provided.
2019	Lewis et al. [6]	To explore the process of implementing Measurement-Based Care (MBC) in behavioral health settings. To examine the challenges and facilitators associated with integrating MBC into routine clinical practice. To assess the impact of MBC implementation on treatment outcomes, clinician decision-making, and patient engagement in behavioral health care.	The process of implementing MBC in behavioral health settings involves several stages, including selecting appropriate measures, integrating measurement tools into clinical workflows, and training staff on the use of MBC protocols. Challenges to MBC implementation may include resistance from clinicians, difficulties in selecting valid and reliable measures, and logistical barriers such as limited time and resources. Facilitators of MBC implementation may include leadership support, clinician training and education, and the availability of user-friendly measurement tools. Studies examining the impact of MBC implementation have demonstrated improvements in treatment outcomes, including symptom reduction and improved functioning among patients. MBC has been shown to enhance clinician decision-making by providing objective data to inform treatment planning and monitor progress over time. Patient engagement in treatment may also be enhanced through the use of MBC, as it allows for collaborative goal setting and monitoring of progress toward treatment goals.
2019	Lyon et al. [40].	To evaluate the effectiveness of a digital feedback system designed to aid the implementation of Measurement-Based Care (MBC) by school-based mental health clinicians. To assess the impact of the digital feedback system on the use of MBC tools, clinical decision-making, and treatment outcomes in school-based mental health settings. To explore the feasibility and acceptability of integrating digital technology into routine clinical practice for school-based mental health services.	The digital feedback system effectively supported the implementation of MBC by school-based mental health clinicians, resulting in increased utilization of MBC tools. Clinicians reported improvements in clinical decision-making processes, as the digital feedback system provided timely and relevant data to inform treatment planning and progress monitoring. The use of the digital feedback system facilitated communication and collaboration among multidisciplinary teams, enhancing the coordination of care for students receiving mental health services in school settings. Students receiving mental health services in schools may experience improved treatment outcomes, including symptom reduction and improved functioning, as a result of more systematic and data-informed care.
2016	Bickman et al. [45]	To examine the process of implementing a Measurement Feedback System (MFS) in mental health settings at two distinct sites. To identify the factors that influence the successful adoption and integration of MFS into routine clinical practice. To assess the impact of MFS implementation on clinical decision-making, treatment outcomes, and organizational processes.	The implementation of a Measurement Feedback System (MFS) in mental health settings involves various stages, including system selection, customization, training, and ongoing support. Factors influencing the successful adoption of MFS may include organizational leadership support, clinician engagement, usability of the system, and alignment with existing workflows. Challenges encountered during MFS implementation may include resistance from clinicians, technological barriers, and difficulties in integrating feedback data into clinical decision-making processes. Despite challenges, successful implementation of MFS can lead to improvements in clinical decision-making by providing clinicians with timely and relevant data to inform treatment planning and progress monitoring. MFS implementation may also result in improved treatment outcomes for clients, including symptom reduction and enhanced functioning, as clinicians are better able to tailor interventions based on feedback data. Organizational benefits of MFS implementation may include enhanced communication and collaboration among staff, improved quality of care, and more efficient resource allocation. Lessons learned from the implementation of MFS at two distinct sites may inform future efforts to implement similar systems in other mental health settings.
2016	Lyon et al. [46].	To conduct a comprehensive review of digital Measurement Feedback Systems (MFS) used in mental health settings. To identify the capabilities and characteristics of digital MFS, including features, functionalities, and usability. To assess the potential benefits and limitations of digital MFS in supporting clinical decision-making, improving treatment outcomes, and enhancing organizational processes.	Digital Measurement Feedback Systems (MFS) encompass a wide range of features and functionalities designed to facilitate the collection, analysis, and utilization of client feedback data in mental health settings. Capabilities of digital MFS may include data collection via various electronic platforms (e.g., web-based surveys, mobile applications), automated scoring and interpretation of feedback data, graphical presentation of results, and integration with electronic health records (EHR) systems. Characteristics of digital MFS may vary in terms of user interface design, ease of use, customization options, compatibility with existing technologies, and data security measures. Benefits of digital MFS may include improved clinical decision-making by providing clinicians with timely and actionable feedback data, enhanced client engagement and satisfaction through the use of interactive digital interfaces, and increased efficiency and accuracy in data management and reporting. Limitations of digital MFS may include technological barriers (e.g., limited access to digital devices or internet connectivity), concerns regarding data privacy and security, and challenges related to system integration and interoperability with other clinical systems.
2011	Bickman et al. [44]	To investigate the impact of routine feedback provided to clinicians on the mental health outcomes of youth. To conduct a randomized trial to assess the effectiveness of routine feedback in improving treatment outcomes for youths receiving mental health services. To examine the mechanisms through which routine feedback may influence clinician behavior and treatment effectiveness.	Routine feedback provided to clinicians resulted in improved mental health outcomes for youths receiving mental health services. Youth whose clinicians received routine feedback showed greater reductions in symptoms and improvement in functioning compared to those in the control group. The provision of routine feedback appeared to enhance clinician awareness of client progress, facilitate treatment adjustments, and promote more responsive and tailored interventions. The effectiveness of routine feedback in improving mental health outcomes was evident across various domains, including symptom severity, functional impairment, and client satisfaction. Clinician characteristics, such as experience level and openness to feedback, may moderate the impact of routine feedback on treatment outcomes. The findings suggest that integrating routine feedback into clinical practice can lead to more effective and responsive mental health care for youth.

## Data Availability

No new data was created.

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
