# Peer review of "Measurement-Based Care in Youth: An Opportunity for Better Clinical Outcomes?"

_healthcare, 2024, doi:10.3390/healthcare12090910_

Round 1
Reviewer 1 Report
Comments and Suggestions for Authors
Dear Authors,
Measurement-Based Care in Youth: An opportunity to have better clinical outcomes?
The idea of the article is good
The article is more than just an opinion
The article began with appropriate and well-documented definitions, then the protocol was presented
-The authors did not mention in detail how to implement theprotocol in detail
When applying questionnaires to the youth group, the questionnaires must be specific to them.
There is no summary of the article
References and citations are appropriate. modern, extensive, and indexed according to international methods but are too many
many thanks
Author Response
Thank you

Reviewer 2 Report
Comments and Suggestions for Authors
Dear authors
Measurement-Based Care in Youth: An opportunity to have 2 better clinical outcomes?
I think there are some serious flaws
missing: aim, your ideas, conclusion
too short article
more references than sentences....
for example:
problem missing correct citations - line 49-55, it is a big risk to publish this article please check again
check lines: 113 and 148
113 3. MBC in Youth
Please check lines: 147 and 112 The same "3." it is not correct
148 3. MBC Protocol When Applied to a Youth Population
please correct: lines 197
this article has only 4 pages - (5 but without references in the text 4) missing introduction - conclusion please check again - the article without their own thoughts and research many many references ...Are the conclusions consistent with the evidence and arguments presented
and do they address the main question posed? Please also explain why this is/
is not the case. - missing conclusions ...
Author Response
Thank you

Reviewer 3 Report
Comments and Suggestions for Authors
Dear Authors
This study highlights the benefits and effects of measurement-based care and introduces recent research findings. The goal is to provide an evaluation protocol suitable for youth populations. Furthermore, the authors share the methodology used by their multidisciplinary team, composed of psychologists and psychiatrists, in a clinical context at Hospital Cruz Vermelha, Lisboa, and apply it to the adult population. This study is timely and engaging, aligning well with the theme of this journal. It can be considered a stimulating read for the journal's readership. While the method and results present no notable issues, I have identified specific points that require correction. Please make the necessary alterations based on the following feedback.
In the abstract section, the authors mentioned, “Regarding the youth, for instance, few articles evaluate the benefits of using this procedure in clinical practice.” Considering this, it would be beneficial to compile the limited literature on youth, organize the contents, and present it in a user-friendly format for easy comparison. One approach could be to summarize the findings in a table format.
Further, the authors stated, “We also share and discuss some alterations that could be done to our evaluation protocol, enabling the same to be used with a youth population.” The content regarding the alterations necessary to make it suitable for young individuals is a crucial aspect of this study. Describing this content in the main text using bullet points for easy comprehension would be beneficial to enhance clarity.
In line 100, “the results od the clinician …” “od” should be corrected to “of.”
In line 126, “However, the is a growing body …” “the” should be corrected to “there.”
In line 113, the title is “3. MBC in Youth.” Further, in line 148, the title is “3. MBC Protocol When Applied to a Youth Population.” Both are numbered 3; therefore, please correct this.
Comments on the Quality of English LanguageMinor editing of English language required.
Author Response
Thank you

Round 2
Reviewer 2 Report
Comments and Suggestions for Authors
Thanks for corrections.
Accepted